# Topical Percutaneous Drug Delivery for Allergic Diseases: A Novel Strategy for Site-Directed Pharmacologic Modulation

**DOI:** 10.3390/pharmaceutics17070867

**Published:** 2025-07-02

**Authors:** Mitsuhiro Kamimura, Hiroaki Todo, Kenji Sugibayashi, Koichiro Asano

**Affiliations:** 1Department of Respiratory Medicine, National Hospital Organization Disaster Medical Center, Tokyo 190-0014, Japan; 2Faculty of Pharmacy and Pharmaceutical Sciences, Josai University, Saitama 350-0295, Japan; ht-todo@josai.ac.jp (H.T.); sugib@josai.ac.jp (K.S.); 3Division of Pulmonary Medicine, Department of Medicine, Tokai University School of Medicine, Isehara 259-1193, Japan; koasano@gmail.com

**Keywords:** cutaneous administration, topical administration, allergic conjunctivitis, allergic rhinitis, cough, asthma, drug targeting

## Abstract

Topical percutaneous drug delivery has recently emerged as a novel strategy for the treatment of allergic diseases, offering targeted drug delivery to mucosal tissues adjacent to the skin. Unlike conventional topical approaches that act on the skin surface or mucosal membranes, topical percutaneous drug delivery enables non-invasive pharmacologic modulation of deeper structures such as the conjunctiva, nasal mucosa, and trachea. This review explores the rationale, pharmacokinetic foundation, clinical data, and future prospects of transdermal therapy in allergic conjunctivitis, allergic rhinitis, and asthma-related cough. In allergic conjunctivitis, eyelid-based transdermal delivery of antihistamines such as diphenhydramine and epinastine has shown rapid and long-lasting symptom relief, with epinastine cream recently approved in Japan following a randomized controlled trial (RCT) demonstrating its efficacy. Preclinical and clinical pharmacokinetic studies support the eyelid’s unique permeability and sustained drug release profile, reinforcing its utility as a delivery site for ocular therapies. In allergic rhinitis, diphenhydramine application to the nasal ala demonstrated symptomatic improvement in patients intolerant to intranasal therapies, though anatomical separation from the inflamed turbinates may limit consistent efficacy. Similarly, cervical tracheal application of steroids and antihistamines has shown potential benefit in asthma-related cough, especially for patients refractory to inhaled treatments, despite anatomical and depth-related limitations. Overall, site-specific anatomy, skin permeability, and disease localization are critical factors in determining therapeutic outcomes. While trans-eyelid therapy is supported by robust data, studies on the nasal ala and trachea remain limited to small-scale pilot trials. No major adverse events have been reported with nasal or tracheal application, but eyelid sensitivity requires formulation caution. To validate this promising modality, further RCTs, pharmacokinetic analyses, and formulation optimization are warranted. Topical percutaneous drug delivery holds potential as a non-invasive, site-directed alternative for managing allergic diseases beyond dermatologic indications.

## 1. Introduction

Allergic diseases such as allergic rhinitis, allergic conjunctivitis, and asthma are among the most prevalent chronic conditions globally [1]. These disorders are characterized by hypersensitive immune responses to environmental allergens, leading to symptoms such as sneezing, nasal congestion, ocular pruritus, cough, and wheezing [2,3,4]. Topical therapy has traditionally been employed in the management of allergic diseases by delivering medications directly to the target tissues via specialized routes: nasal sprays for allergic rhinitis, eye drops for allergic conjunctivitis, and inhaled corticosteroids (ICS) and bronchodilators for asthma, which are designed to deliver pharmacological agents directly to the affected tissue, thereby reducing systemic exposure and associated adverse events (AEs) [5,6,7]. However, even a combination of systemic administration via oral medications and conventional topical therapies does not always result in adequate symptom control. In some cases, the occurrence of local adverse effects from topical formulations prevents the use of sufficient drug dosages.

Topical percutaneous drug delivery targets deeper, non-dermal structures adjacent to the skin, distinguishing it from conventional topical approaches that act primarily on the epidermis, dermis, and hypodermis. This technique avoids direct application to mucosal surfaces and systemic administration, aiming to improve symptom control while minimizing local irritation. Preliminary pilot studies have demonstrated that transdermal application of diphenhydramine cream (Restamin, Daiichi Sankyo Co., Ltd., Tokyo, Japan) can provide symptomatic relief in allergic conjunctivitis [8], rhinitis [9], and asthma-related cough [10], offering a potentially valuable adjunct to conventional therapies. While promising, these results are preliminary, and efficacy has not been confirmed in all cases. Although the initial findings are encouraging, further research is needed to clarify the role of transdermal therapy in the management of allergic diseases. Of particular note is the development and approval of epinastine cream (Alesion Eyelid Cream, Santen Pharmaceutical Co., Ltd., Osaka, Japan), a second-generation antihistamine cream for allergic conjunctivitis [11]. The successful application of transdermal antihistamines in allergic conjunctivitis may support the broader application of transdermal therapy in the treatment of allergic rhinitis and asthma.

This review explores the rationale, mechanisms, existing clinical data, and future directions of percutaneous drug delivery in allergic disease management.

## 2. Pharmacologic Basis of Transdermal Therapy

Traditionally, topical percutaneous drug delivery has been primarily employed for the treatment of localized pain in subcutaneous tissues, such as joints and muscles, using non-steroidal anti-inflammatory drug (NSAID) patches or ointments. After penetrating the stratum corneum, drug molecules reach the dermis, where a dense capillary network typically absorbs the majority into systemic circulation. Only a small fraction of the drug escapes vascular absorption and continues to diffuse into deeper tissues [12].

Nonetheless, several studies have demonstrated that certain topically applied agents, particularly NSAIDs, can achieve higher concentrations in muscle [13] or joint tissues [14,15] than in plasma, indicating effective localized drug delivery. NSAIDs such as diclofenac have been shown to penetrate approximately 3–4 mm beneath the skin surface, where their concentrations in muscle and joint tissues can exceed those measured in the systemic circulation [16]. In a rat model, it was reported that under passive diffusion conditions, most of the drug was delivered within the first 2–3 mm of tissue depth, with negligible penetration beyond 5 mm unless chemical enhancers or assisted techniques such as iontophoresis were employed [17].

In addition to their well-established antipruritic effects, H1-antihistamines are known to possess anti-inflammatory [18] and analgesic properties [19,20]. For example, topical application of diphenhydramine cream to the skin overlying painful osteoarthritic joints has been reported to exert significant analgesic effects [20], suggesting its ability to reach subcutaneous target tissues. This pharmacodynamic profile raises the possibility that transdermally delivered antihistamines may exert therapeutic effects in non-dermal allergic diseases located near the skin surface.

Based on these findings, the concept of utilizing transdermal drug delivery to modulate allergic inflammation in anatomical sites immediately adjacent to the skin—such as the conjunctiva, nasal mucosa, and trachea—presents a compelling therapeutic rationale. While conventional topical therapies for allergic disease have largely targeted the dermis, as in atopic dermatitis or urticaria, expanding the reach of these agents to submucosal and periepithelial structures via transdermal techniques warrants further investigation and may offer a novel modality in allergic disease management.

## 3. Application to Allergic Conjunctivitis

Allergic conjunctivitis, characterized by itching, tearing, redness, and swelling of the conjunctiva, is commonly treated with topical eye drops containing antihistamines or mast cell stabilizers [4]. However, eye drops have several limitations. First, ocular pharmacokinetics of eye drops are suboptimal due to rapid drug clearance from the ocular surface through tear turnover, blinking, and drainage via the nasolacrimal duct, resulting in low bioavailability—typically only 5–10% [4,21]. To achieve sufficient therapeutic effects, multiple daily instillations—often two to four times—are commonly required. This frequent dosing regimen can be particularly challenging for pediatric patients, leading to reduced adherence and inconsistent treatment outcomes [21,22]. Additionally, practical difficulties such as poor instillation technique and limited motor coordination, especially in young children and the elderly, may further compromise the effectiveness of topical ocular therapies [21,22].

Ultrasound biomicroscopy has revealed that the full thickness of the lower eyelid—from the epidermis to the palpebral conjunctiva—can be estimated at approximately 3.35 mm, based on the summation of layer-specific anatomical measurements [23]. Given the proximity of the skin surface to the conjunctival side, and the limitations associated with topical instillation, transdermal delivery via the eyelid was devised as an alternative approach.

The first clinical application of trans-eyelid drug delivery in humans was reported in 1999, involving the use of a calcium-based petrolatum ointment applied to the lower eyelid skin for the treatment of dry eye disease [24]. In a double-masked, controlled trial, patients exhibited significant improvements in subjective symptoms, ocular surface staining, and blink rate over a three-month period. Notably, the use of fluorescein-labeled petrolatum confirmed that the drug vehicle reached the tear film via percutaneous migration and remained detectable for up to six hours, indicating a sustained-release mechanism. The next clinical report utilizing this delivery approach was a small pilot study investigating the use of 1% diphenhydramine ointment for the treatment of allergic conjunctivitis [8]. Diphenhydramine was administered at a dose of 0.2 mg per eyelid twice daily, resulting in a total daily dose of 0.8 mg. The study enrolled seven participants, five of whom completed the protocol. All seven participants demonstrated clinical improvement, with most reporting rapid symptom relief within three minutes of application and effects lasting between five and 24 h, notably exceeding the duration typically observed with conventional eye drops. AEs were observed in four patients. Two patients discontinued treatment due to local AEs: one developed palpebral edema and pain, and the other developed ocular erythema and pruritus associated with pain. These symptoms resolved spontaneously following treatment discontinuation. The remaining two patients continued treatment despite experiencing mild AEs, including transient blurred vision during the first week and mild eyelid tingling. Importantly, no increases in intraocular pressure or changes in visual acuity were observed in any participants during the study period, including those who discontinued treatment. These studies demonstrated that trans-eyelid application resulted in longer-lasting therapeutic effects than traditional eye drops, likely due to slow release from skin and subcutaneous tissues acting as a drug reservoir.

The pharmacokinetic advantages of trans-eyelid drug delivery demonstrated in clinical settings have been further explored in preclinical studies. Animal experiments have shown that the eyelid skin exhibits significantly greater permeability than other anatomical sites, with studies reporting 6-fold and 11-fold higher transdermal absorption of diclofenac and tranilast, respectively, compared to abdominal skin [25]. In addition to its permeability, the eyelid functions as a local drug depot, wherein both the skin and subcutaneous tissues retain lipophilic agents and gradually release them into adjacent ocular structures [25]. Several pharmacokinetic investigations have confirmed these findings across different animal models. In a rabbit model, once-daily application of a stick-type ketotifen formulation to the eyelid skin achieved therapeutic drug concentrations in the conjunctiva [26]. In hairless rats, topical administration of tranilast to the lower eyelid resulted in significantly prolonged ocular drug exposure relative to eye drops and intravenous injection. Pharmacokinetic analysis revealed that mean residence times in the conjunctiva and eyeball were extended by up to 8.4-fold and 4.5-fold, respectively, following eyelid application [27]. These findings support the feasibility of trans-eyelid drug delivery as a sustained-release approach for chronic ophthalmic conditions. The efficacy of this delivery route has also been validated using epinastine, a long-acting antihistamine, in two distinct animal models. In rabbits, topical application of epinastine to the eyelid skin maintained therapeutic concentrations in the conjunctiva for up to 24 h, indicating effective ocular delivery and supporting its use as a prolonged-release formulation [28]. In guinea pigs, a single application of 0.5% epinastine cream significantly inhibited conjunctival vascular permeability and scratching behaviors induced by histamine and ovalbumin, with therapeutic effects persisting for 24 h and exceeding those achieved with epinastine eye drops [29]. Collectively, these findings highlight trans-eyelid epinastine therapy as a promising long-acting strategy for the treatment of allergic conjunctivitis.

Building on these findings, a phase 3, double-masked, randomized, intra-patient controlled trial was conducted to evaluate the safety and efficacy of 0.5% epinastine hydrochloride cream applied to the eyelids in adults with a history of seasonal allergic conjunctivitis [11]. Thirty asymptomatic patients (60 eyes) were enrolled, and the left and right eyes of each participant were randomized to receive either the epinastine cream (approximately 30 mg) or a placebo applied to the upper and lower eyelid skin. A conjunctival allergen challenge was performed 24 h after application to assess ocular allergic responses. The epinastine-treated eyes demonstrated significantly lower scores for ocular itching and conjunctival hyperemia compared to placebo-treated eyes, with therapeutic effects lasting at least 24 h. The formulation was well tolerated, with no treatment-related AEs reported. These cumulative findings confirm the prophylactic efficacy and safety of trans-eyelid epinastine delivery and led to its clinical approval in Japan. The cream is now available as a once-daily, non-invasive therapeutic option and represents a promising alternative for the management of allergic conjunctivitis.

## 4. Application to Allergic Rhinitis

Allergic rhinitis affects a significant proportion of the global population and frequently coexists with asthma [2,3,30], with reports indicating that up to 67% of asthma patients in Japan also suffer from allergic rhinitis [30]. Conventional treatment options include oral antihistamines, intranasal corticosteroids, intranasal antihistamines, and leukotriene receptor antagonists [2,3]. However, intranasal therapies are frequently associated with local mucosal AEs, including nasal burning, irritation, pain, epistaxis, throat discomfort, and unpleasant taste [9,31,32,33]. Among these, epistaxis is reported as the most common AE of intranasal medications [32,33,34]. These local side AEs are primarily attributed to mucosal dryness, epithelial thinning, and repeated mechanical or chemical irritation. Benzalkonium chloride, a common preservative in nasal formulations, is recognized as a potent mucosal irritant. Its application has been linked to immediate discomfort, such as nasal pain and stinging upon contact with the nasal mucosa [35]. Chronic use of benzalkonium chloride-containing sprays may further cause cumulative epithelial injury, impaired ciliary function, and rebound congestion consistent with rhinitis medicamentosa [36]. Importantly, the incidence of epistaxis appears to increase with treatment duration—for example, from 3.2% after two weeks of intranasal antihistamine use to 19–25% after 12 months; intranasal corticosteroids exhibit similar trends with a reported 17–23% incidence after one year [33]. These findings indicate the cumulative burden of intranasal therapies on the nasal mucosa.

In this context, alternative delivery strategies that reduce mucosal toxicity while preserving therapeutic efficacy are warranted. Given the relatively thin skin and anatomical proximity to nasal structures, the nasal ala represents a potentially feasible and less invasive site for transcutaneous drug delivery.

To date, only a single small pilot study has been conducted [9], and no animal experiments have been performed due to the lack of an animal model with a nasal structure comparable to that of humans. In this study, transdermal application of diphenhydramine cream to the nasal ala was evaluated in a cohort of 10 patients with allergic rhinitis and asthma, all of whom had previously demonstrated clinical responsiveness to intranasal ketotifen. The treatment involved applying approximately 0.7 mg of diphenhydramine cream to both nasal alae twice daily for two weeks. Clinical effectiveness was observed in 50% (5 out of 10) of patients, while an additional 30% (3 patients) reported mild improvement. Although the median time to onset was slower than that of intranasal ketotifen (30 min vs. 10 min), the duration of effect was comparable between the two treatments, with both lasting approximately 5 h. Importantly, no local AEs were observed—even among individuals who had previously discontinued intranasal therapies due to mucosal irritation such as nasal pain or epistaxis. Furthermore, four patients who had experienced nasal irritation from intranasal sprays rated the transdermal approach more favorably. This observation suggests a compelling use case for patients who are intolerant of intranasal formulations. Additionally, two patients, whose asthma symptoms were exacerbated by postnasal drip, experienced concurrent improvement in both postnasal drip and asthma control following treatment, suggesting that amelioration of upper airway inflammation may contribute to better management of lower airway disease [37].

In allergic rhinitis, the primary sites of allergic inflammation are the inferior and middle turbinates [38,39,40], where the extensive respiratory mucosa and rich vascularization promote vigorous allergic responses. In contrast, involvement of the nasal ala mucosa in allergic inflammation is relatively limited compared to the turbinates. Consequently, transdermal drug delivery to the nasal ala cannot be expected to achieve the uniform efficacy observed with intranasal ketotifen. Nevertheless, transdermal application of diphenhydramine cream to the nasal ala demonstrated clinically meaningful benefits in a subset of patients. These findings suggest that despite its limitations, the nasal ala may serve as an alternative, non-invasive delivery site worthy of further exploration, particularly for patients unable to tolerate conventional intranasal therapies.

## 5. Application to Asthma-Related Cough

Cough is a common manifestation of asthma and is especially prominent in phenotypes such as cough-predominant asthma and cough-variant asthma, where it represents the primary or sole clinical symptom [41]. ICS and bronchodilators remain the cornerstone of therapy for all asthma phenotypes [7], as the primary site of inflammation in both cough-variant asthma and cough-predominant asthma is the bronchial airway which is consistent with the pathophysiological characteristics of classical asthma [41,42]. Several studies indicated that airway inflammation in asthma extends beyond the bronchi to involve the trachea. High-magnification bronchovideoscopic examinations have revealed marked hypervascularity of the tracheal mucosa in patients with newly diagnosed asthma, suggesting active tracheal inflammation [43]. Moreover, mechanical cough challenge tests—such as tracheal vibration or compression over the cervical trachea—have been shown to reliably provoke cough during active disease phases, with a notable reduction in cough sensitivity as disease activity resolves [44]. These findings collectively support the trachea as an active site of inflammation in asthma. A study demonstrated that patients with cough-variant asthma or cough-predominant asthma who were refractory to conventional dry powder inhaler therapy experienced improved symptom control with nebulized corticosteroids, likely due to more effective drug deposition in the tracheal region [45]. These observations suggest the potential clinical value of therapeutic strategies that specifically target tracheal inflammation in the management of asthma-related cough, particularly in cases resistant to standard inhaled regimens.

ICS, either alone or in combination with inhaled bronchodilators, represent the mainstay of local therapy for asthma management [7]. In addition, oral agents such as leukotriene receptor antagonists, and to a lesser extent, antihistamines have been reported to be beneficial [46,47]. However, the use of ICS may sometimes be limited due to oropharyngeal AEs such as hoarseness, oral thrush, or stomatitis, resulting in suboptimal disease control owing to necessary restrictions on the inhalation dose [48,49].

Since the trachea is part of the lower respiratory tract and can be directly palpated through the skin, it represents a unique anatomical site for non-invasive therapeutic approaches. Although substantial interindividual variability exists, largely influenced by factors such as body habitus and body mass index, reported measurements indicate that the mean skin-to-trachea distance is approximately 9.2 ± 1.9 mm in adults [50] and ranges from 4.6 mm to 6.2 mm in pediatric populations depending on age [51]. While the trachea lies deeper than anatomical sites such as the eyelid or the nasal ala, its relatively superficial location compared to other internal structures suggests the potential feasibility of topical percutaneous drug delivery in this region.

To date, only limited study exists regarding transdermal drug delivery targeting the trachea, consisting of a single animal study [52] and a small-scale clinical pilot investigation [10]. The feasibility of delivering prednisolone to the trachea via transdermal application to the cervical skin was evaluated in an animal model using iontophoresis to enhance drug penetration [52]. In this experimental setting, topical application of prednisolone succinate to the cervical skin of rats alone resulted in measurable drug levels within the tracheal tissue, demonstrating that transdermal delivery to the lower airway is achievable. Furthermore, the application of iontophoretic stimulation produced approximately a 12-fold increase in tracheal drug concentration compared to passive diffusion, highlighting the substantial potential of this technique to optimize local drug delivery.

In a clinical pilot study [10], 28 patients with bronchial asthma, cough-variant asthma, or cough-predominant asthma were evaluated for the effect of transdermal application of steroid ointment (either mometasone furoate or betamethasone valerate) and diphenhydramine to the cervical trachea skin once or twice daily for up to three months. The amount applied per session was approximately 150 μg for steroids and 2–3 mg for diphenhydramine. Of these patients, 11 (39.3%) demonstrated a reduction in cough symptoms, with three achieving complete resolution. Among the same cohort, diphenhydramine ointment was additionally tested in 14 patients, resulting in five responders (35.7%), all of whom had also responded to steroid therapy; notably, one patient showed improvement with diphenhydramine application despite previously failing oral olopatadine treatment. Although subjective in nature, the onset of action in effective cases was typically within one hour, and the duration of effect lasted approximately three to five hours. While the overall response rate was lower compared to transdermal applications at the eyelid or nasal ala, one possible contributing factor is anatomical limitation: the cervical trachea represents approximately 40% of the entire tracheal length, and the transdermal approach can cover only the anterior aspect of the trachea. Thus, this therapy may not sufficiently address inflammation involving the posterior or more distal regions of the airway. Nevertheless, these findings suggest that transdermal therapy targeting the cervical trachea may serve as a third route of anti-inflammatory treatment for airway inflammation, complementing inhalation and systemic administration.

## 6. Comparative Assessment

Beyond allergic conjunctivitis, the application of transdermal diphenhydramine for other inflammatory conditions remains largely unexplored. Among the available pilot studies, diphenhydramine is the only agent that has been consistently tested across allergic conjunctivitis, rhinitis, and airway inflammation. Accordingly, a comparative assessment focusing on these pilot studies utilizing diphenhydramine offers a unique opportunity to evaluate differential therapeutic outcomes across these conditions.

### 6.1. Clinical Effectiveness of Topical Diphenhydramine Therapy on Each Site

Clinical efficacy appears to be greatest in allergic conjunctivitis, followed by rhinitis, with the lowest efficacy observed in airway inflammation. This disparity may be attributed to the proportion of inflamed tissue directly adjacent to the transdermal application site relative to the total surface area of the affected organ (Figure 1, Table 1).

In eyelid application, a relatively large proportion of the inflamed conjunctival surface may be covered. In contrast, in the case of the trachea, the cervical tracheal skin application can access only approximately less than 40% of the tracheal length and is limited to the anterior surface. Additionally, the skin-to-trachea distance varies considerably among individuals, particularly due to differences in subcutaneous fat thickness, further limiting the consistency of drug delivery. When viewed in the context of the entire airway, the area covered by cervical transdermal application remains exceedingly small.

The nasal ala constitutes the lateral boundary of the external nasal vestibule, whereas the inferior and middle turbinates are located deeper within the nasal cavity [53,54]. Among the three turbinates, the inferior turbinates are the largest and extend prominently into the nasal airway from their bony origin. Although its anterior end lies relatively close to the nasal ala, it is not situated directly beneath it. Instead, it is positioned posterior to the ala’s attachment point on the lateral nasal wall, with its anterior tip projecting further away. The middle turbinates lie superior and posterior to the inferior turbinates, creating an even greater anatomical distance from the nasal ala. Importantly, the inferior and middle turbinates—key sites of allergic inflammation in rhinitis [38,39,40]—are not in immediate proximity to the nasal ala. This spatial separation may partially account for the variability observed in the clinical effectiveness of topical treatments applied to the area. Moreover, considerable interindividual and racial differences in the thickness and surface area of the nasal ala [55] may further influence drug absorption and, consequently, therapeutic response. These anatomical and physiological factors make consistent efficacy across populations difficult to ensure.

### 6.2. AEs of Topical Diphenhydramine Therapy and Site-Dependent Sensitivity

In pilot studies, no AEs were reported following diphenhydramine ointment application to the nasal ala or the cervical tracheal region [9,10]. In contrast, multiple AEs were observed with eyelid application, including eyelid edema, pain, pruritus, ocular discomfort, and transient blurred vision [8]. While these reactions were generally non-severe and either resolved spontaneously with continued use or subsided after discontinuation, their frequency was markedly higher than that associated with application to the nasal ala or cervical trachea.

Notably, no AEs were reported with eyelid application of epinastine ointment [11]. Although speculative, differences in excipient composition represent one possible factor contributing to this discrepancy. The diphenhydramine cream contains known irritants such as sodium lauryl sulfate and ethanol, both of which are absent from the epinastine cream. Detergents like sodium lauryl sulfate are well-documented cutaneous irritants [56,57] and may have contributed to the observed adverse reactions. Contact dermatitis is a rare but recognized AE of topical antihistamines. Reports have described cases of allergic contact dermatitis caused by diphenhydramine ointment, with potential cross-reactivity to related compounds [58]. Eyelid contact dermatitis is frequently reported due to the exceptional sensitivity of periorbital skin to topical agents, including cosmetics and ophthalmic medications [59], indicating the vulnerability of periorbital skin.

Skin properties vary significantly across anatomical sites. Compared to truncal skin, facial skin is characterized by a thinner stratum corneum, higher surface hydration, and diminished barrier function. The neck exhibits skin characteristics comparable to those of the cheek, with similarly diminished barrier function and increased sensitivity to irritation relative to the forearm [60]. Within the face, further site-specific variability is evident: the eyelid possesses the thinnest stratum corneum, the lowest sebum content, and the highest hydration levels, yet exhibits the weakest barrier integrity [61]. Eyelid skin is also notably soft [62] and highly permeable, with transdermal penetration rates significantly exceeding those of abdominal skin in animal models—by up to 6-fold for diclofenac and 11-fold for tranilast [25]. These unique properties of the eyelid contribute to its heightened vulnerability to topical agents.

Given these unique anatomical and physiological features, special caution is warranted when applying topical formulations to the eyelid. Although no AEs were recorded with short-term (2-week) use on the nasal ala, the potential for delayed AEs with long-term application cannot be excluded. This concern is supported by evidence showing that the incidence of AEs, such as epistaxis, increases with prolonged use of intranasal antihistamines or corticosteroids [33]. Consequently, the long-term safety of transdermal antihistamine therapy—particularly in sensitive regions such as the eyelid—remains uncertain and warrants further investigation.

**Table 1 pharmaceutics-17-00867-t001:** Comparative anatomical and pharmacological characteristics of the three transdermal application sites targeted in topical percutaneous drug delivery.

	Eyelid	Nasal Ala	Pretracheal Skin
target	conjunctiva	nasal cavity, mainly turbinates	trachea
Thickness (mm)	3.35 [23]	5–7 *	4.6–9.2 [50,51]
location of the target organ	directly beneath	close to turbinates but not directly beneath	directly beneath
Anatomical coverage of the target organ by the application site	full	partial	partial
barrier function	weak [61,62]	moderate [61]	moderate [60] **
drug absorption	high [25]	ND	ND
sensitivity to irritation	high [59]	ND	high [60] ***
topical percutaneous diphenhydramine delivery: efficacy	high [8]	partly [9]	less partly [10]
topical percutaneous diphenhydramine delivery: AEs	frequent [8]	none [9]	none [10]

ND: no data. * Estimated from CT images by the author (M.K.); no published data on nasal ala thickness are available. ** Higher than eyelid, close to cheek; slightly lower than forearm. *** Less than eyelid, similar to cheek; more than forearm.

## 7. Future Research Directions

Except for trans-eyelid epinastine therapy, which has undergone a randomized controlled trial (RCT) and gained regulatory approval in Japan, most studies to date have been small-scale pilot investigations involving limited sample sizes and subjective outcome measures. To establish the clinical validity and generalizability of this therapeutic modality, rigorous double-blind, placebo-controlled RCTs with adequate statistical power are needed.

In allergic rhinitis, no animal model adequately replicates this site. Consequently, the progression of nasal ala–based therapies must rely entirely on carefully designed human clinical studies. Future trials should incorporate objective endpoints such as eosinophil counts, fractional exhaled nitric oxide (FeNO), nasal airflow metrics, and validated symptom scales to ensure scientific rigor and reproducibility.

In contrast, the eyelid has emerged as a highly promising site for transdermal drug delivery. Beyond allergic conjunctivitis, its application has also been investigated in glaucoma therapy by using pilocarpine. In rat models, trans-eyelid administration of pilocarpine has been shown to produce significantly higher concentrations in the conjunctiva and intraocular tissues than conventional eye drops, with miotic effects maintained for up to 8 h [63]. Subsequent pharmacokinetic modeling, based on data from rabbit studies, has predicted that in humans, peak concentrations in the iris and aqueous humor would be achieved within 1–2 h following eyelid application, with therapeutic effects lasting approximately 4.5 h [64]. These findings highlight the eyelid as a broadly applicable and non-invasive route for targeted ocular drug delivery as shown in trans-eyelid antihistamines. This expanding therapeutic scope further highlights the need for detailed pharmacokinetic studies to clarify drug distribution patterns and establish optimized dosing strategies for various ophthalmic and periocular conditions.

Moreover, advances in dosage forms—such as semisolid bases (gels, creams), transdermal patches, and films—combined with emerging enhancement technologies including microneedles, iontophoresis, electroporation, sonophoresis, thermal actuation, and magnetophoresis, have the potential to improve drug permeability and bioavailability. These active methods employ physical stimuli (e.g., electric fields, ultrasound, heat, or magnetic forces) to transiently disrupt the skin barrier or propel drug molecules into deeper tissues, thereby enabling localized or systemic effects beyond the limits of passive diffusion [65]. Comparative studies assessing a range of drugs—including newer-generation antihistamines, steroids, and NSAIDs—will be essential to identify the most suitable compounds for transdermal therapy. Furthermore, given the substantial interindividual variation in skin thickness, hydration, and permeability across anatomical sites and ethnic groups, personalized delivery strategies tailored to patient-specific skin characteristics may optimize both efficacy and tolerability.

## 8. Conclusions

Topical percutaneous drug delivery represents a promising and novel frontier in the treatment of allergic diseases. Notably, skin-based drug delivery may hold therapeutic value not only for dermatologic conditions—such as atopic dermatitis and urticaria—but also for diseases of the nasal cavity, respiratory tract, and ocular surface. This review outlined its potential applications in allergic conjunctivitis, rhinitis, and asthma-related cough. Evidence from eyelid application demonstrates its feasibility, tolerability, and potential clinical utility. Preliminary studies involving nasal ala and cervical tracheal application suggest similar therapeutic benefits, although the current evidence remains limited. Nevertheless, these site-specific strategies may offer valuable, patient-centered alternatives for individuals who are intolerant to conventional intranasal or inhaled treatments. To validate this emerging strategy, well-designed RCTs and pharmacokinetic studies, including active transdermal delivery systems, are warranted.

## Figures and Tables

**Figure 1 pharmaceutics-17-00867-f001:**
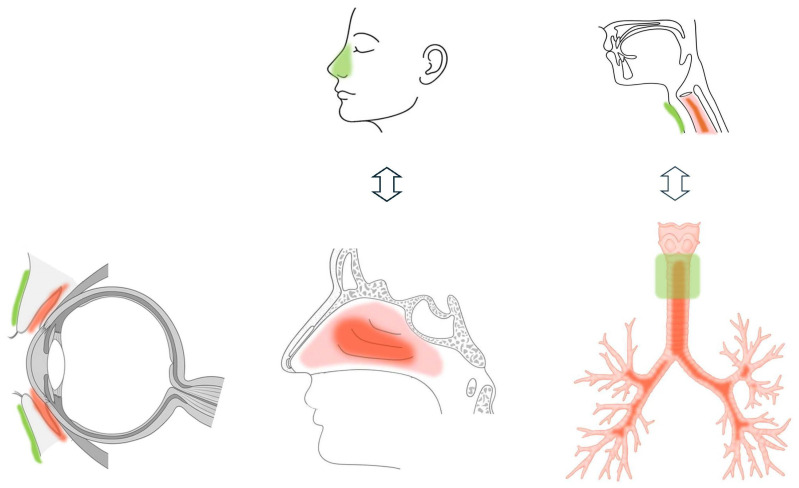
The figure shows a comparison of the anatomical positioning of diphenhydramine application sites (green) relative to the sites of inflammation (red) in three conditions: allergic conjunctivitis, allergic rhinitis, and asthma-related cough. In eyelid application, the application area substantially overlaps with the inflamed conjunctiva, supporting high clinical efficacy. In nasal ala application, the site is near the inflamed turbinates but does not directly overlie them. In cervical tracheal application, although the drug is applied directly above the inflamed trachea, the accessible surface area is anatomically limited. These differences in spatial proximity and coverage ratio between the application site and the inflamed tissue likely explain the gradient of therapeutic effectiveness observed across conditions (eyelid > nasal ala > trachea).

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
