# Peer review of "Topical Percutaneous Drug Delivery for Allergic Diseases: A Novel Strategy for Site-Directed Pharmacologic Modulation"

_pharmaceutics, 2025, doi:10.3390/pharmaceutics17070867_

Round 1
Reviewer 1 Report
Comments and Suggestions for Authors
The authors present a well-written and very interesting review into the use of topical therapies for the treatment of allergic diseases.
I have the following general comments and questions:
- The authors write (Lines 56-57):
“Transdermal drug delivery targets deeper, non-dermal structures adjacent to the skin, distinguishing it from 56 conventional topical approaches that act primarily on the skin surface.”
However, topical therapy is not limited for action “on” the skin surface. The majority of drugs will act in the viable epidermis and dermis. Therefore, the authors should be more precise and correct the text. More generally it would be useful for them to define the differences between their use of the term “transdermal delivery” and the more conventional use to describe drugs that enter the systemic circulation.
- In the next section (Lines 71-80), the authors describe the “transdermal” delivery of NSAIDS for the treatment of localized pain and inflammation; it is important to distinguish between treatment of these conditions in muscle and in joints. It is difficult to understand how local delivery would result in increased levels within the synovial regions without passing through the systemic circulation. Results seen in small rodents may not translate to treatments in humans.
- The “Comparative Assessment” section is an interesting summary and relates the issues for delivery to the anatomy. This is a key point and the authors should consider adding figures showing the anatomical regions covered in the review. It is clear that “transdermal delivery” across the eyelid probably has more potential than the other routes of administration for future applications. However, it will be important to find the most appropriate animal models.
Reviewer 2 Report
Comments and Suggestions for Authors
Transdermal Topical Therapy for Allergic Diseases: A Novel Strategy for Site-Directed Pharmacologic Modulation
This manuscript provides a concise overview of transdermal antihistamine formulations as a novel approach for managing allergic conditions.
Although the topic is interesting for this journal, several issues have been identified during the review process, as outlined below.
Comment 1- Most of the manuscript is a compilation of clinical trial reports describing transdermal delivery of antihistamines in allergic diseases. While the authors present clinical outcomes from selected studies, the review lacks a foundational analysis of the anatomical and physiological characteristics of the three key skin regions discussed in the introduction, namely, the eyelid, nasal ala, and cervical trachea, which are critical for understanding the feasibility and limitations of transdermal drug delivery in these areas.
Moreover, the manuscript does not provide an overview of the main types of transdermal delivery systems available, nor does it describe the composition of the ointments used in the cited clinical studies. Information on excipient types and formulation strategies is crucial, as it directly influences drug penetration and therapeutic efficacy. Although some brief remarks on formulation appear near the end of the manuscript, this topic deserves a more prominent and structured discussion, particularly in relation to available penetration-enhancement strategies.
In addition, the authors do not address how these formulations can be characterized or what types of in vitro and in vivo studies would be necessary to evaluate drug permeation and delivery to the target tissue.
Trans-eyelid drug delivery, for example, is a well-established route with numerous formulation types beyond ointments, including gels, solutions, and nano-based systems, many of which are approved and commercialized worldwide, not just in Japan. However, the authors provide little discussion on these alternatives and miss the opportunity to critically compare them.
Finally, the manuscript lacks the authors' own perspective on which transdermal strategies may be most promising. The conclusion briefly suggests the potential for various transdermal systems but falls short of discussing them in any depth or offering an expert opinion on future directions. Overall, this study is very concise, but it would benefit from a more detailed discussion of the advances brought by transdermal formulations in enhancing the effectiveness of antiallergic treatments. Expanding on how these systems improve drug delivery, patient compliance, or therapeutic outcomes would provide valuable context and strengthen the overall impact of the manuscript.
Comment 2- It is recommended to include schematic diagrams or summary tables to help readers better understand the differences between the three transdermal application sites discussed in the paper. The manuscript would greatly benefit from visual elements, such as figures or illustrations adapted from relevant studies, that depict the various classes of transdermal delivery systems used on the eyelids, nasal ala, and cervical trachea. For example, additional diagrams could include simplified anatomical cross-sections highlighting differences in skin thickness, vascularization, and proximity to target tissues. Further illustrations could show the types of formulations employed (e.g., ointments, gels, creams, or iontophoretic systems), their penetration pathways, and mechanisms of action. These visuals would enhance clarity and allow for a more intuitive comparison, particularly given the anatomical and functional differences between these regions.
Round 2
Reviewer 2 Report
Comments and Suggestions for Authors
The authors have satisfactorily clarified all the issues I raised. However, I am unable to find the figure mentioned in their cover letter.